# Evolution of Microstructure and Texture in Laboratory- and Industrial-Scaled Production of Automotive Al-Sheets

**DOI:** 10.3390/ma13020469

**Published:** 2020-01-19

**Authors:** Jakob Grasserbauer, Irmgard Weißensteiner, Georg Falkinger, Stefan Mitsche, Peter J. Uggowitzer, Stefan Pogatscher

**Affiliations:** 1Christian Doppler Laboratory for Advanced Aluminum Alloys, Montanuniversitaet Leoben, A-8700 Leoben, Austria; irmgard.weissensteiner@unileoben.ac.at (I.W.); stefan.pogatscher@unileoben.ac.at (S.P.); 2AMAG Rolling GmbH, A-5282 Ranshofen, Austria; georg.falkinger@amag.at; 3Institute for Electron Microscopy, Graz University of Technology, Steyrergasse 17, A-8010 Graz, Austria; stefan.mitsche@felmi-zfe.at; 4Chair of Nonferrous Metallurgy, Department Metallurgy, Montanuniversitaet Leoben, A-8700 Leoben, Austria; peter.uggowitzer@unileoben.ac.at

**Keywords:** aluminum alloys, AlMgMn alloys, AlMgSi alloys, texture evolution, electron backscattered diffraction

## Abstract

With the rising importance of aluminum sheets for automotive applications, the influence of microstructure and texture on mechanical properties and on forming behavior has gained re-increased interest in recent years. This paper provides an introduction to the topic and demonstrates the evolution of microstructure and texture in the standard alloys EN AW-5182 and EN AW-6016 for different processing scales. Moreover, strategies for texture and microstructure characterization of automotive Al-sheets are discussed. As the development of alloys or processes usually starts in laboratory facilities, the transferability to the industrial scale of the results thereof is studied. A detailed analysis of the entire processing chain shows good conformity of careful laboratory production with the industrial production concerning microstructure as well as qualitative and quantitative texture evolution for EN AW-5182. While comparable grain sizes can be achieved in final annealed sheets of EN AW-6016, quantitative discrepancies in texture occur between the different production scales for some sample states. The results are discussed in light of the basics of plasticity and recrystallisation including the effect of solutes, primary phases, and secondary phases in the alloys.

## 1. Introduction

In recent decades, the use of aluminum components has increased, particularly in the automotive industry, as the associated weight reduction leads to a reduction in CO_2_ emissions and can further promote the use of electromobility. In order to meet the increasing demands on safety standards and mechanical properties, a deep understanding of the mechanisms underlying the various application-relevant properties is required [1,2,3,4]. Alloys for automotive body applications often belong to one of the two aluminum classes, 5xxx (AlMg(Mn)) or 6xxx (AlMgSi). The Mg and Mn containing 5xxx Al-alloys fall into the category of non-age-hardenable Al-alloys and show good formability, high strength, corrosion resistance, and weldability. The mechanical properties are controlled by the Mg content and the degree of cold forming. Owing to age-hardenability of 6xxx Al-alloys, controlled thermo-mechanical processing is crucial in these alloys. The various possible strength levels, together with high fracture toughness and good corrosion resistance, open a wide field of applications [3]. Over the last decades, the influence of crystal orientation on the mechanical properties and on the forming ability of the Al-components has been intensively investigated and has recently gained eminent interest with the rising importance of Al-sheet materials for automotive applications [5,6,7].

As far as the orientation distribution of the grains within forming operations is concerned, the texture of a material describes the entirety of the crystallographic orientations. However, textured materials are commonly understood to exhibit intensified orientations that are characteristic of the strain conditions of previous forming operations. The progress in measurement techniques over the last decades provides a variety of possible methods for texture characterization. In general, the most prominent techniques are X-ray diffraction (XRD) and electron backscattered diffraction (EBSD), both having beneficial aspects. X-ray diffraction texture measurements show improved statistics when compared with those of EBSD, because of larger sampled areas. However, with the lack of microstructural information in the X-ray measurements (only macrotexture measurements are possible), EBSD is superior in terms of achieving meaningful results concerning microstructure and texture simultaneously (microtexture). In any case, the usage of these techniques provides reliable information concerning texture analysis and the suitability of XRD or EBSD depends at least on the sample processing conditions and the information sought [8].

Owing to the face-centered cubic crystal structure of Al, typical deformation structures occur in the rolling process. The basic deformation mechanism of dislocation slip, which is also relevant in other crystal structures, occurs on twelve independent {111}<110> slip systems and generally results in structural features of different dimensions, namely, cells or subgrains, deformation bands, and shear bands [9,10,11]. In addition to these structures, the influence of alloying elements and their interaction with precipitation is crucial. The automotive alloys EN AW-5182 and EN AW-6016 examined in this study exhibit some typical characteristics, the detailed discussion of which we consider to be essential.

The investigation of the microstructure and texture of Al-sheets first requires a very precise description of the parameters regarding the position within the sheet thickness. The different strain conditions within the sheet result in a variety of microstructural features. The resulting texture components can be completely different, which also affects the final sheet properties. While high-strained near-surface layers often show distinct shear components, the microstructure in the center of a sheet often suffers more typical plane-strain conditions [12,13]. A stronger formation of substructure and thus more stored energy will be found closer to the surface of the sheets. Another feature typical for highly strained areas is a 25°–40° inclining of the subgrain dislocation networks and band-like structures to the rolling direction (RD) [14].

The complex interactions of all these phenomena can lead to a large diversity of annealed sheet grain structures. As far as EN AW-5182 is concerned, the chemical composition facilitates the formation of shear bands owing to the slightly reduced stacking fault energy and the enhanced solute drag. In addition, Mn-containing dispersoids limit grain growth. For EN AW-6016, the systematic control of second-phase particles is crucial as the age-hardening phases can readily form in earlier stages of the processing route and affect the microstructure of the final sheet. These basic mechanisms form the backbone for understanding microstructure development in such alloys [6,15].

Intense research in the field of texture and the related properties of Al-sheets lead to a common understanding of the typical mechanisms for rolling- and recrystallization-texture evolution [16,17,18,19]. Typical cold-rolling processes for standard fcc-alloys form components such as Brass, Copper, and S (see Figure 1), also referred to as *β*-fiber [20,21]. These stable orientations are prominent for the active octahedral slip systems at cold-rolling temperatures; however, hot-rolling conditions including elevated temperatures above 320 °C tend to activate non-octahedral slip. Publications on this topic show in this case the stabilization of Brass, but also Cube orientation {001}<100> [22]. 

Typical recrystallization textures in Al are dominated by the Cube component, which arises from Cube nuclei in the deformation zones [23,24,25,26,27,28]. Earlier theories of texture evolution stated the controversy between oriented nucleation and oriented growth. Concerning Cube grains, one can find coincidences of both theories, as they nucleate from preserved Cube bands and show favorable growth conditions in to typical deformation texture components [29,30,31,32]. Major influences on Cube texture are given by the degree of cold-rolling and annealing temperatures as well as solute elements [33]. However, especially in EN AW-5182, the Mg in solid solution can readily reduce the dominance of the cube texture by influencing the dislocation cell formation, especially at lower strains [5,34]. 

In EN AW-6016, second phase particles show potential to modify occurring textures by various mechanisms. In general, the influence of particles on the final texture can be divided into two main mechanisms. Small, closely spaced particles with sizes below 500 nm are able to increase the Smith–Zener pinning on grain boundaries, and thus delay recrystallization [35]. Depending on the time of particle formation during processing, the resulting texture may have an enhanced or retained Cube component [36,37]. Additionally, existing particles with a size > 1 µm act as favorable nucleation sites. The mechanism of particle stimulated nucleation (PSN) is based on heavier deformation substructures in the vicinity of these particles, which provide higher driving force for recrystallization. Investigations indicate the formation of rather random texture components or at least a reduction of the predominant Cube orientation in the vicinity of these particles [38,39,40].

Exemplary textures of annealed EN AW-5182 and EN AW-6016 are shown in Figure 1. While Figure 1b indicates typical recrystallization texture dominated by the Cube component for the EN AW-6016; the orientation distribution for the EN AW-5182 appears more random and weaker.

As new alloys or process changes are often only implemented in laboratory facilities, the further development of aluminum alloys is very often based on small series production. However, difficulties can arise in the transferability of the results to industrial scale, and the comparability of laboratory and industrial products must always be guaranteed. The first difficulties in the casting process can already be observed in the typical processing of wrought aluminum alloys.

In particular, the Al-series AlMg(Mn) (5xxx) and AlMgSi (6xxx) both require precise knowledge of the parameters in casting, as micro- and macro-segregations have to be controlled. As the distribution of the alloying elements as primary phases or dissolved elements can have a significant influence on the subsequent deformation process, controlled cooling conditions upon casting are indispensable [41]. In the case of age-hardenable Al-alloys, a homogenization treatment should generally be carried out to homogeneously distribute the hardening phase-forming elements and counteract segregations [15,42,43,44]. In addition, the rolling geometries can influence the overall texture and microstructure development. The most important factors in this case are the diameter and friction of the rolls. As these two parameters differ for laboratory and industrial rolling mills, the final textures may show a significant inconsistency [9].

In this study, we evaluate the microstructure and texture evolution in the two Al-alloys EN AW-5182 and EN AW-6016 on different processing scales. Particular attention is paid to individual processing steps and the underlying changes in microstructure and texture in laboratory scale. The results will be discussed in the light of well-established theories and will demonstrate the need for careful consideration of the differences between laboratory and industrially produced Al-sheets.

## 2. Materials and Methods 

The alloys EN AW-5182 and EN AW-6016 were studied. The chemical composition is given in Table 1.

Sample production is based on a conventional production process of 5xxx and 6xxx alloys, as shown schematically in Figure 2 [45,46]. The usual industrial processing parameters [4,15] were used for the industrially scaled path, the aluminum sheets produced by Austria Metall AG (AMAG) rolling GmbH. For the laboratory production route, the complete processing of both alloys—consisting of alloy production, casting, rolling and annealing—was carried out in the small laboratory facilities described below (see also the work of [47] for further details).

Casting: The ingot material for both alloys was provided by AMAG rolling GmbH company, Ranshofen, Austria. Initially, the ingot material was remelted in a resistance-heated furnace (Nabertherm K20/13/S, Lilienthal, Germany), degassed using an impeller with Ar gas, and finally casted into laboratory-scaled molds [45]. After milling to block sizes of 80 × 80 × 40 mm^3^, samples of EN AW-6016 were homogenized in an air-circulated furnace for 10 h at 565 °C with subsequent air cooling, while EN AW-5182 remained untreated.

Rolling: For the laboratory scaled rolling process, a small-scale laboratory rolling mill with roll diameter of 250 mm was used. Both alloys were hot- rolled and cold-rolled (HR and CR), including an intermediate soft annealing (IA) heat treatment at 370 °C in between the cold-rolling passes. Maximum reductions per pass were 2 mm for hot-rolling and 1 mm for cold-rolling. From the initial thickness of 40 mm, the EN AW-5182 was hot-rolled to a thickness of about 2.5 mm, and subsequently cold-rolled to 1.5 mm before intermediate annealing. EN AW-6016 was hot-rolled to 7.3 mm, cold-rolled to 3.1 mm, and intermediate annealed. For the IA heat treatment, an air-circulated furnace (Nabertherm N60/85SHA, Lilienthal, Germany) was used. Relatively low heating and cooling rates are intended to mimic large scale processing of coils in a batch furnace. The final sheet thickness was specified to be 1.20 mm and 1.25 mm for the 5182 and 6016 alloys, respectively. The dimensions were achievable within a tolerance of 0.03 mm in thickness.

Heat treatment: The final heat treatments, meaning soft annealing (SA) for EN AW-5182 and solution annealing (also SA) for EN AW-6016, were performed in a salt bath for 5 min at temperatures of 500 °C and 530 °C, respectively, to mimic an industrial continuous heat treatment line. Subsequently, the samples were water quenched.

As the aim of this work was to acquire a sound knowledge of texture and microstructure evolution, samples were examined at different stages of the production process. In addition to the information given in Figure 2, cast and homogenized microstructures were prepared by means of metallographic methods, but only briefly investigated for the quality of the casting and homogenization process.

The material of the designated processing steps was further processed by means of standard metallographic preparation in two sample orientations: (1) the cross-section spanned between rolling direction (RD) and normal direction (ND); (2) position at *s* = 0.5, with Equation (1) as metallographic samples in the sheet plane comprising the RD and transverse direction (TD).
(1)$$s=\frac{2D_{t}}{t_{0}},$$
where *D_t_* represents the distance from center and *t*_0_ represents the initial thickness [15].

The sample preparation included cutting, embedding, grinding, and polishing, as well as polishing with an oxide polishing suspension (OPS, Struers) at relatively high contact forces of up to 50 N for at least 18 s. In addition to the OPS-polishing reactant, unperfumed liquid soap was used to improve lubrication during preparation.

Measurements of both alloys required further treatment for sufficient surface quality. For EBSD, electropolishing (EP) was performed on a Struers Lectro-Pol 5 electropolishing unit with electrolyte A2 (provided by Struers) for 4–8 s at 36 V and 10 °C.

Microstructure and texture were investigated using scanning electron microscope (SEM) (JEOL 7200F FEG-SEM, Tokyo, Japan) equipped with an EBSD-measurement system (Nordlys Nano detector, Oxford Instruments, Abingdon, UK). For qualitative analysis, backscattered electron (BSE) images of the prepared sections were also collected. The EBSD-measurements were done using a 70° pre-tilt specimen holder and accelerating voltage of 20 kV. Acquisition and analysis of the diffraction patterns required a large variation in parameters, as deformed and annealed sample states were measured.

### EBSD Data Processing

The post processing and evaluation of EBSD data were done by means of the Matlab based toolbox MTEX 5.2.beta3 [48].

Grain size analysis: The linear intercept length (LIL) is calculated for the data from EBSD measurements. For the grain reconstruction, a tolerance angle of 5° was chosen and larger areas of connected, not-indexed pixels (e.g., primary phases) were excluded from the dataset. The rest of the EBSD data were smoothed by applying a half-quadratic filter [49].

Texture Analysis: The raw data were rotated manually and centered by means of stereographic projections of the discrete orientations. For a sound representation of a materials’ texture, the orientation information of a minimum number of 1000 [50,51] to 3000 spatially independent grains [52,53] is required. In the present study, at least 3000 grains were analyzed for all sample conditions of EN AW-5182. For EN AW-6016, the analysis was more challenging. The large grain size of the as-cast condition leads to large, similarly oriented areas after hot-rolling and even after the first cold-rolling cycles. However, at least 7.5 mm^2^ of the hot-rolled and at least 1.5 mm^2^ of the cold-rolled material were analyzed by EBSD with step sizes depending on the amount of stored deformation. Note that XRD measurements of the hot-rolled material did not lead to better results, although a beam of 1 cm^2^ was applied to measure samples of 23 × 23 mm^2^.

To quantify changes in texture and to compare it to the XRD data, orientation distribution functions (ODF) were calculated from the EBSD data in the mtex toolbox. The ODF best represents the entirety of discrete orientations when combining the calculation via a direct kernel density estimation and the kernel density estimation via Fourier series. The semi-automatic half width selection algorithm was applied on discrete orientations as well as on the mean orientation (weighted by area) from “orientation clusters”, which are grains defined by a misorientation threshold of 2°. The half widths for the kernel density estimation were confined between 6° and 10° for the individual orientations (the resulting ODF is shown in Figure 3a) and between 6° and 11.5° for the orientation clusters (ODF in Figure 3b). The ODFs calculated via both methods (without any symmetrization) were finally averaged (Figure 3c), which, in the present study, reduces the under-smoothing of the data and still conserves the maxima of the data sets. This procedure also leads to the lowest deviation of the volume fractions of ideal orientations derived from the ODF to the volume fractions directly extracted from the EBSD data (Figure 3d).

Statistical methods to correctly determine the accuracy of the ODF calculated from measured pole figures were shown in the work of [54], and error estimations on ODFs calculated from individual orientations are presented in the works of [53] or [55]. The works of [53,54] have in common an artificial variation of the input data and a subsequent comparison of the mean of the input data to estimate the relative uncertainty. As the EBSD data in the present study have a high number of random orientations that cannot be assigned to ideal orientations, we did not perturbate simulated textures to compare the outcome of the ODF, but compared the evaluated volume fractions with the directly evaluated ones from EBSD data of very large datasets. Then, 3000 ODF calculations on parts of the dataset (50,000 individual orientations and 2500 mean orientations of areas with max. 2° misorientation) were performed to evaluate the standard deviation between these ODFs and the ODF calculated from the whole dataset. The maximum standard deviation of the intensity was 7 × 10^−7^, which proves a reliable representation of the individual orientations by means of the ODF. The standard deviation correlates with the intensity; however, the relative error is highest in areas with low intensity.

The texture of the material was quantified by evaluating the respective volume fractions of the ideal components (orientations listed in Table 2) for including the symmetric equivalents) from the ODF. Therefore, the tolerance angle of the ideal orientations was chosen as 10°, which comprises orientations with a certain angular spread on the one hand and almost completely avoids overlapping between the ideal components on the other hand, as the minimum angle between them is 19.38°.

## 3. Results

The following sections present the evolution of microstructure and texture in 5182 and 6016 Al-alloys on the different processing scales. The microstructural evolution is discussed based on micrographs as well as on grain size analyses. The change of texture during processing is presented in the form of pole figures based on ODF calculations and volume fractions of ideal texture components extracted from these ODFs.

### 3.1. Microstructure and Grain Parameters

#### 3.1.1. Microstructure Evolution of EN AW-5182

The micrographs in Figure 4a–f show the microstructure of laboratory produced alloy 5182 for the three different process steps: hot-rolling, cold-rolling (after IA), and soft annealing. For better recognition of all structural details, both backscattered electron (BSE) images and band-contrast (BC) images are shown.

First of all, the upcoming aspects of deformation in the hot-rolled state can be recognized in Figure 4a,b. In the laboratory process, the material was hot-rolled to 2.5 mm from an initial thickness of 40 mm. The corresponding microstructure shows slightly elongated grains in the rolling direction, an increasing dislocation network at the grain boundaries (red arrow Figure 4b) with typical inclining of cell-band boundaries (slip bands) to RD (green arrow Figure 4b), as well as a linear alignment of fragmented primary phases and secondary phases (blue arrow in Figure 4a; secondary phases are hardly visible at this magnification). Black areas result from a preferential dissolution during preparation, which does not affect the quality of the statements. The BC image given in Figure 4b indicates some grain boundaries, which enables first estimations of the grain sizes in the process states. 

Figure 4c,d show the microstructure after the final cold-rolling pass (according to CR after IA). The cold-rolling degree of 20% with a final sheet-thickness of 1.20 mm leads to a typical rolling structure, which, in this special case, appears somewhat comparable to the hot-rolling microstructure. On closer look, it reveals a more pronounced subgrain formation and a tendency for reduced grain size in the CR state. Again, the band-contrast images give a good overview of the grade of deformation and the associated grain size, although shadowing of primary phases occurs. 

Figure 4e,f show the soft annealed microstructure of the laboratory sheet. Owing to the orientation contrast and the absence of subgrain structures, the different grains are clearly recognizable already in the BSE image. It is striking that a relatively coarse-grained microstructure appears in the final state, even though some grains still show an elongation in the rolling direction, and thus have no balanced aspect ratio. The BC image also reveals preferential attack of specific grain orientations by electrolyte during EP.

The comparison of the BSE and BC images highlights the importance of the usage of EBSD for microstructural investigations, especially in the deformed states.

Additional Energy-dispersive X-ray spectroscopy (EDS) analysis of particles identified the formation of (primary-)Al–(Fe,Mn) constituents and small (secondary-) particles—dispersoids with higher Fe and Mn concentrations compared with the matrix. During processing, the primary phase particles align to the rolling direction and easily fragment owing to the stress conditions during rolling. A more detailed analysis of dispersoids composition is not concerned in this study.

More information on the microstructure is given by the grain linear intercept length illustrated in Table 3, deduced from large area EBSD mappings. In order to allow an initial comparison, the parameters for the industrially processed Al-sheets are added. 

Table 3 compares the linear intercept lengths of the industrially and laboratory produced sheets of EN AW-5182. Particularly noticeable are the differences in the mean linear intercept length in the individual process steps. Especially, the results after the individual cold-rolling steps open a discussion about the deformation energies introduced in industry and laboratory production, as the linear intercept length for the lab-scale products shows a significant increase compared with the industrial material. In addition to the large differences in LIL after the cold-rolling stages, however, a common tendency to form a very similar grain size after the annealing stages can be observed. 

#### 3.1.2. Microstructure Evolution of EN AW-6016

The micrographs in Figure 5a–f display the results for EN AW-6016. For the hot-rolled condition (Figure 5a,b), one pronounced aspect is the extremely large grain size. With grains potentially exceeding the image- and scan-size limits, the assessment of texture and microstructural evolution is affected by larger errors in this processing state. The orientation contrast in the BSE images shows less and heavier deformed subgrain structures within the grains. Preferential attack of electrolyte in EP and shadowing by primary phases again is recognizable in the BC images.

The cold-rolled microstructure in Figure 5c,d shows noticeable subgrain development and heavily deformed microstructure. Moreover, a band-like deformation structure with directional alignment to RD of the sub-cells is observable. A comparison to EN AW-5182 in Figure 5c,d indicates stronger deformation structures within the grains of EN AW-6016, which is because of the higher cold-rolling degree for EN AW-6016.

In the solution annealed EN AW-6016, BSE and BC images show good accordance regarding the qualitative microstructure and occurring features. Overall, the microstructure shows polygonal, almost equiaxed grains with only a slight remaining stretch in RD.

A comparison of the LIL deduced from large area EBSD mappings for laboratory and industrial products is given in Table 4. As mentioned above, the hot-rolled images in Figure 5 indicate grains far in excess of the scanned range, meaning that the LIL values for hot-rolled conditions should not be taken into account. Nevertheless, as the grain sizes greatly exceed the expectations on both scales, these results should not be connected to differences in lab- and industry-processing. Starting from IA, the data exhibit better accordance. The following cold-rolling steps suggest differences in the underlying deformation states for laboratory and industry, as the disagreement in LIL is more pronounced. Concerning the linear intercept length in the recrystallized sample states IA and SA, both show a much better accordance. Particularly, the final solution annealed samples exhibit comparable LIL.

### 3.2. Texture Analysis

#### 3.2.1. Texture Evolution in EN AW-5182

A description of the results regarding the orientation distribution during the process steps in the alloy EN AW-5182 is made based on the illustrations shown in Figure 6. First, in Figure 6a–f, pole figures calculated from ODF data are shown for the processing states HR, CR2, and SA (laboratory: Figure 6a,c,e; industry; Figure 6b,d,f). A more comprehensive quantification of the texture evolution for specific components is shown in Figure 6g–j, where the diagrams depict either rolling or recrystallization texture components. 

Starting with hot-rolled conditions, pole figures Figure 6a,b show the typical *β*-fiber formation in both laboratory and industry samples, although the total intensities are not equal. Owing to the elevated temperatures, the entire rolling texture is rather weak, but still shows pronounced Brass and S components. Comparing laboratory and industry samples, the pole figures show good accordance of the orientations occurring. During further processing, cold-rolling leads to intensified β-fiber components, especially in the industrially produced samples. Therefore, the pole figures in Figure 6c,d illustrate typical orientation distribution for cold-rolled fcc-materials. The results given in Figure 6e,f clarify the obvious differences in rolling and recrystallization texture formation. The β-fiber components are significantly reduced and the volume fraction of typical orientations of recrystallized grains (Cube, Q, P) increases; however, the overall intensity of recrystallization texture is weak for both laboratory and industry samples. 

The representation of proportions of some specific texture components, given in Figure 6g–j, enables an easier comparison of the entire process. The evolution of rolling texture components in Figure 6g,h verifies the presumptions based on the pole figures. Typical β-fiber components dominate the texture after rolling processes and are weakened by recrystallization (heat treatments). The good agreement of the texture components, not only qualitatively, but for most of the sample states also quantitatively, is emphasized with these diagrams.

The components of the recrystallization texture in Figure 6i,j also confirm theoretical considerations from the literature [12]. Strengthening of these components occurs with heat treatments and recrystallization, while they develop at the expense of the rolling components. Although the evolution of recrystallization components generally still shows conformity for laboratory and industry, the Q component is intensified in laboratory samples throughout the entire process. Additionally, the cube component does not dominate the final sheet structure in contrast to classical recrystallization textures of low alloyed fcc-systems.

#### 3.2.2. Texture Evolution in EN AW-6016

The results on texture evolution in EN AW-6016 are presented in Figure 7a–j. The orientation distribution for hot-rolled sample states shown in the pole figures in Figure 7a,b reveal only a few different orientations. While formation of typical β-fiber is expected during hot-rolling of Al-alloys, both laboratory and industrially hot-rolled samples exhibit higher Cube fractions. The higher amount of deformation energy introduced during cold-rolling tends to form the β-fiber in both laboratory and industry sheets. Figure 7c indicates more typical S and Copper components in the laboratory sample, while the industrial sample (Figure 7d) also shows Brass orientations.

The pole figures of the solution annealed sample states in Figure 7e,f show good accordance for laboratory and industry. Theoretical aspects of texture formation are observable, particularly as Cube component domination is indicated on both scales of sample processing.

The entire texture evolution in Figure 7g–j confirms most of the indications based on the pole figures. While the typical trend of rolling texture components over process is found for the industrial sample, the Brass component in the laboratory sample vanishes for the first cold-rolling process. Although quantitative differences are evident, both industry as well as laboratory follow expected behavior during the rolling process.

The trend of recrystallization texture components for the industrial sample in Figure 7j depicts the classical fcc-texture evolution. The dominating Cube component is accompanied by lower volume fractions of Q, P, and CubeND in the annealed sample states. Laboratory samples show similar behavior for the evolution of recrystallization components, and even though the Q component dominates the final texture of the sheet, the overall texture development is qualitatively and, for specific sample states, also quantitatively comparable for industry and laboratory.

## 4. Discussion

In the following, the comparability of texture and microstructure development in industrial and laboratory samples will be discussed, including some considerations on general texture evolution in Al-alloys.

The microstructure evolution of the EN AW-5182 laboratory-sheets generally confirms the results and theories of previous publications [9]. While typical microstructural features such as subgrain dislocation networks can be observed in both laboratory and industrial scaled processing [14,56], it should be noted that the deformation energy introduced during rolling is generally expected to be lower in the laboratory sheets. A comparison of the hot-rolled microstructure with the industrial samples reveals the absence of a strong deformation substructure on laboratory scale, which can be attributed to reduced rolling forces in the laboratory mill. Subsequent cold-rolling increases the deformation structure, but still, based on the linear intercept length for both CR1 and CR2 given in Table 3, energy input and grain fragmentation are higher in the industrial process. The less fragmentation of grains may also result in the increment of linear intercept length in both CR states, as the elongation of the grains in RD is not suppressed. After intermediate annealing and soft annealing, however, similar results in grain size and microstructure are observed for the EN AW-5182. The differences of about 10% can be attributed to the deformation energy input during rolling as well as to impacts from differences during laboratory and industrial heat treatments.

As far as the influence of particles on the overall microstructure development is concerned, different mechanisms must be considered. As shown in Figure 4a, various types and sizes of particles are present. Coarse (also fragmented), non-shearable particles enhance the misorientation of the grain in their nearest surrounding by accumulation of dislocations, which can result in grain refinement around those particles after annealing. Secondary phases in the nanometer range often hinder the recrystallization process (grain boundary pinning) [9,57]. A comparison of BSE images of laboratory and industrial sheets shows a similar phase distribution owing to sophisticated processing (especially casting). Although potential particles are present in the studied alloys on both scales, there is no clear evidence for specifically related microstructure transformations. Despite this, as both laboratory and industrial sheets exhibit equivalent behavior concerning particle-related phenomena, the overall comparability of the production process with respect to microstructure development shows satisfactory results.

Moreover, regarding texture evolution in EN AW-5182, Figure 6a–j shows good agreement between laboratory and industrial sheets. Figure 6a–d highlight the strong development of β-fiber texture components, most prominently the S and Brass orientations [5,58,59]. Additionally, the comparison of the intensities in the pole figures for laboratory and industry indicates fewer rolling structures in laboratory processing. The quantification of texture components from ODFs over the process in Figure 6g–j also emphasizes the promising results for the comparability of laboratory and industry alloy production in terms of microstructure and texture. 

General minor intensities of the recrystallization texture components, including Cube orientation, suggest randomization of texture to a certain degree (especially in the laboratory alloy). Concerning the strengthening of the Q component during recrystallization, the sometimes-stated origin near to shear bands (reported in some Mg-containing heavily-rolled Al-alloys) could not be verified due to the absence of attendant microstructural features and the rarely emerging Goss component, which can also indicate features of shear during deformation [6,11,37,60]. As the entire texture development does not show any influence of PSN, a minor significance of microstructural changes based on coarse particles can be assumed; however, a more detailed analysis of the primary and secondary phase distribution will nevertheless be the subject of future investigations.

The comparison of the microstructure development of EN AW-6016 in laboratory and industrial samples shows similarities for individual processing steps. Starting from hot-rolling, fragmented grains with enormous elongation show typical characteristics of the rolling process in the laboratory sheets (Figure 5a,b). Additionally, the linear intercept length listed in Table 4 indicates comparable grain sizes for industrial products. As high standard deviations of the linear intercept length in hot-rolled and CR1 sample states emphasize the imbalance of present grains, further discussion of measured grain size is only useful for a qualitative comparison of laboratory and industrial samples beginning at the IA process step.

The hot-rolled microstructure shows the alignment of subgrain structures in specific angles to RD, as well as inhomogeneous distribution of deformation structures over the sampled area. These inhomogeneities arise from the different orientations and sizes of the grains, which causes an unbalanced stress propagation over the whole sample [9]. The higher deformation energy introduced during cold-rolling forms massively distorted grains with elongations in RD in both laboratory and industry. While there are some discrepancies occurring between laboratory and industry for the linear intercept length in HR and CR1, better comparability of the average linear section length for EN AW-6016 can be observed from IA onwards. Possible explanations are inequalities in the rolling process for laboratory and industry regarding forces and roll gap geometries. As the stored energy is also the driving force for recrystallization, intermediate annealing transfers existing differences to annealed sheets as well. Nevertheless, the final sheet microstructure and the linear intercept length show good accordance on laboratory and industrial scale. Originating from good agreements in CR2 state, the final annealing tends to produce comparable microstructures, confirming in particular the comparability of these heat treatments in terms of time and heating rate.

Interesting behavior can be seen in the texture evolution of EN AW-6016 samples. Owing to the already discussed problem of giant grains in the hot-rolled condition, the pole figures depicted in Figure 7a,b indicate highly textured Al-sheets in both laboratory and industry. As this data are based only on the orientations of a few grains, the validity of the plots is strongly restricted; however, laboratory and industrial samples both show the appearance of Cube orientation, while the tendency of β-fiber formation is not clearly noticeable in the evaluated data. As mentioned in the work of [15], enhancement of the Cube component in hot-rolled sheets can occur by post-dynamic recrystallization, which also constitutes a possible process in our laboratory production.

Following the pole figures in Figure 7c,d as well as the texture evolution for rolling components, typical β-fiber formation can be noticed in every rolling process step. The extinction of Brass component in the initial cold-rolled laboratory sample is not comprehensible, as fundamental publications report its occurrence in (cold-)rolled fcc-materials [9,12], though the industrially processed material does adhere to theoretical predictions. Overall, the trends in texture formation are again comparable for laboratory and industry scale production.

Solution annealed samples show typical recrystallization components in Figure 7e,f,i,j [6,7]. The dominant Cube component is accompanied by other stable orientations in recrystallized industrial sheets, namely, Q, P, and CubeND. Laboratory sheets show minor Cube intensities, but more distinct formation of Q component in Figure 7i. The evolution of these specific components does not highlight specific mechanisms concerning texture modifications. The absence of the Goss component and missing microstructural features suggest the absence of shear banding in the present alloys, although slightly intensified P and Q texture components are noticeable in the annealed sheets [57,61,62].

Another possible explanation for the balanced distribution of recrystallization components in Figure 7 may lie in the presence of particles. However, as stated in various publications [35,36,38,39], the PSN connected texture transformation is generally not clearly observable, as in Figure 7e,f or Figure 7i,j. The slightly higher volume fractions of components P, CubeND, and Q suggest that particles capable of PSN may only be present in the laboratory produced sheets. These possible inequalities in particle distribution for laboratory and industry sheets may result from scale effects during thermomechanical processing, which affects the precipitation of the hardening phases, and thus impacts the texture formation. 

As qualitative conformity in microstructure and texture evolution for the EN AW-6016 is found for the entire process, the comparability of laboratory and industrial processes can be endorsed. However, while the microstructure shows good quantitative agreements in the term of final grain size, the quantitative texture development is not comparable for all processing steps.

## 5. Conclusions

This work provides a comprehensive overview on microstructure and texture evolution of the Al-alloys EN AW-5182 and EN AW-6016. Moreover, it investigates the influences of different processing scales and highlights the importance of carefully designed laboratory experiments. The following conclusions can be made:Texture and microstructure evolution throughout the process shows good conformity for EN AW-5182, both qualitatively and quantitatively. Typical rolling and recrystallization behavior of fcc-metals is found.For EN AW-6016, results from laboratory and industrial scale can be compared qualitatively over the process chain, although quantitative differences do occur. Slight differences in both rolling and recrystallization texture components emphasize the necessity of more detailed analysis of some sample states.Final annealed sheets show highly comparable microstructure and texture in both laboratory and industrially produced sheets for both alloys.Although EN AW-5182 and EN AW-6016 both include alloying elements forming primary and secondary phases, the influences of particle stimulated nucleation or intensified Zener-pinning are not clearly observable in either of these alloys.

In general, careful design of laboratory production of automotive aluminum sheets is necessary to study new alloys or processes. However, at least qualitative comparison to industrially processed sheets should be possible.

## Figures and Tables

**Figure 1 materials-13-00469-f001:**
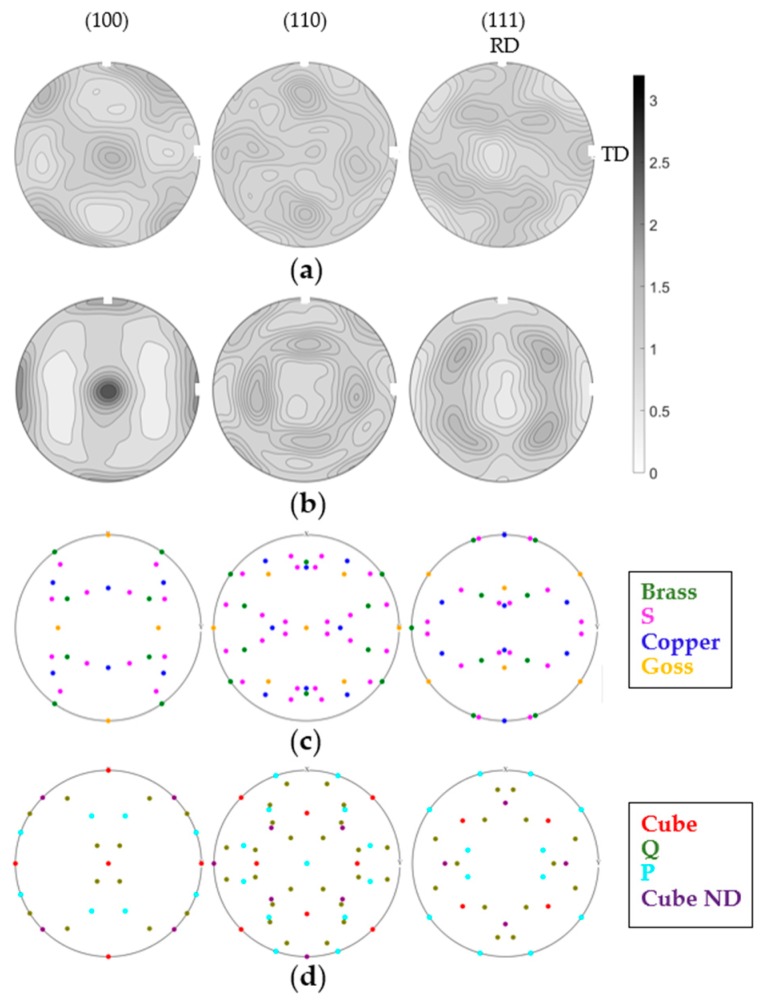
Typical textures of annealed (**a**) EN AW-5182 and (**b**) EN AW-6016; ideal orientations of main (**c**) rolling texture components and (**d**) recrystallization texture components. RD, rolling direction; TD, transverse direction; ND, normal direction.

**Figure 2 materials-13-00469-f002:**
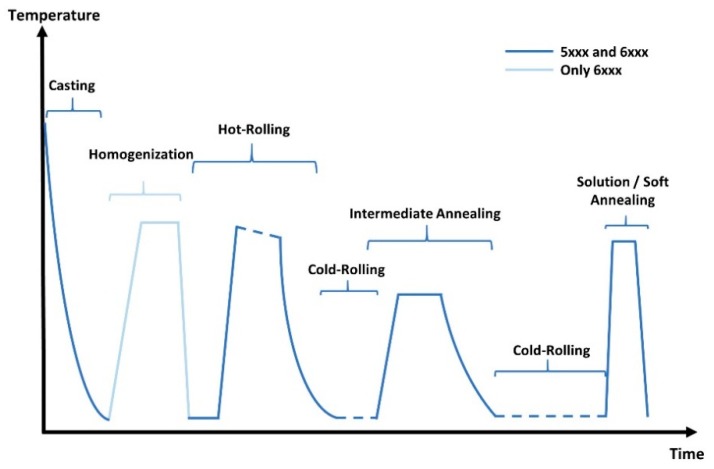
Schematically illustrated process route for EN AW-5182 (no homogenization) and EN AW-6016.

**Figure 3 materials-13-00469-f003:**
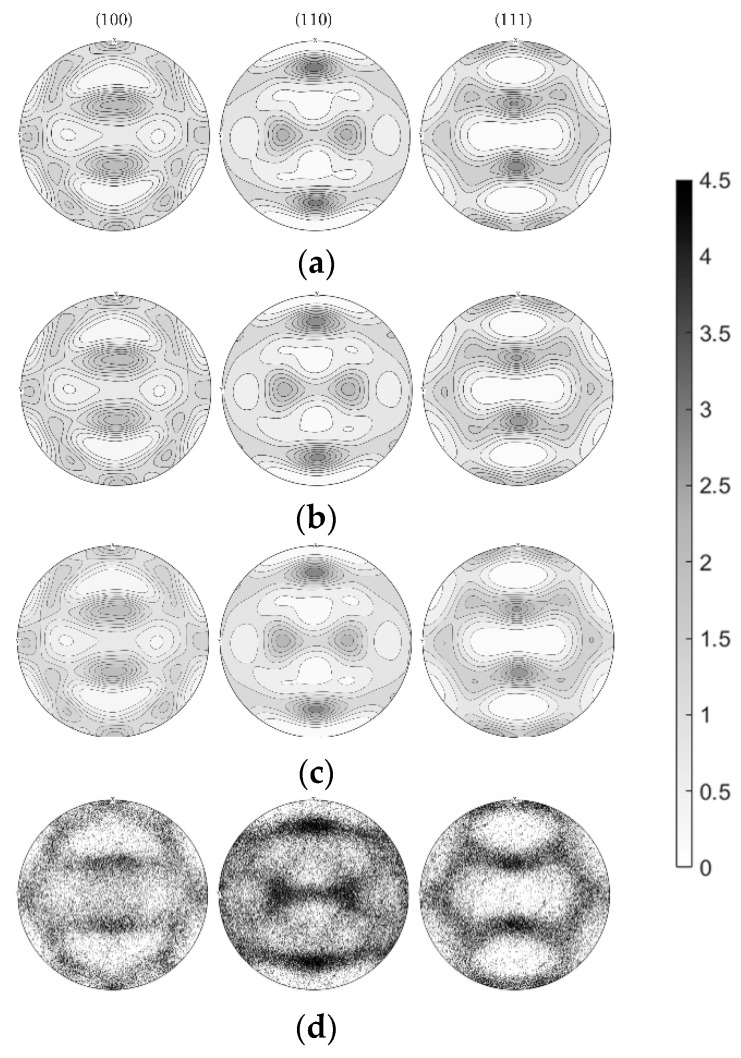
(**a**) Orientation distribution function (ODF) calculated from orientation clusters, (**b**) ODF calculated from individual orientations, and (**c**) the final averaged ODF compared to (**d**) the stereographic projections of the individual orientations.

**Figure 4 materials-13-00469-f004:**
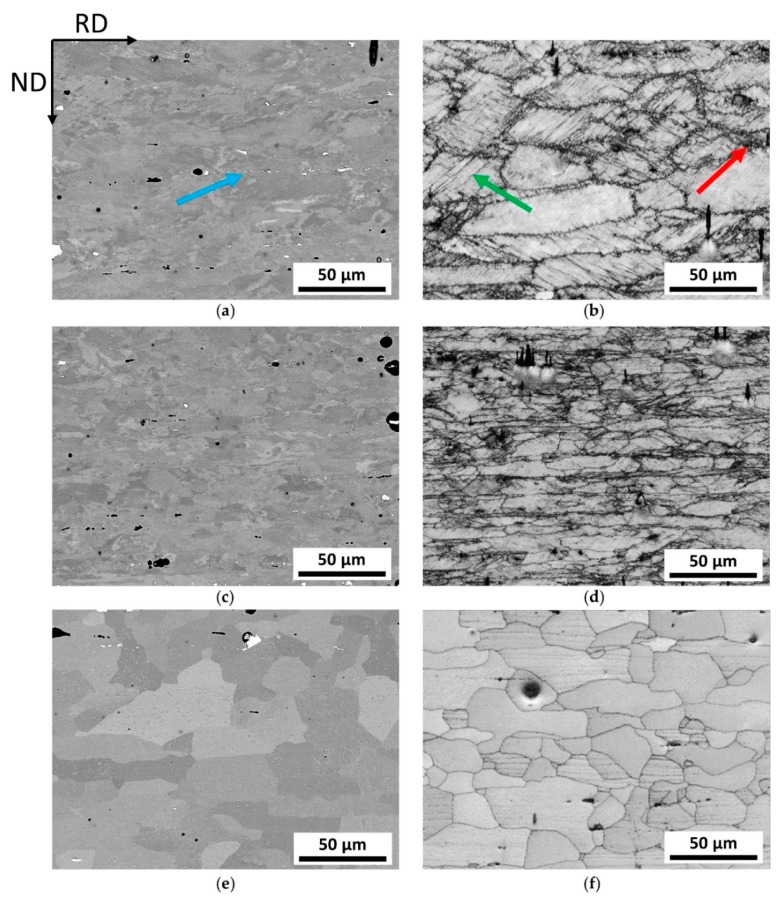
Microstructure evolution of the laboratory processed EN AW-5182 plotted in the TD-plane; (**a**,**c**,**e**) show (**a**) backscattered electron (BSE) images of the hot-rolled, (**c**) cold-rolled (after IA), and (**e**) soft annealed state, respectively; (**b**,**d**,**f**) show (**b**) band contrast maps of the hot-rolled, (**d**) cold-rolled (after intermediate annealing (IA)), and (**f**) soft annealed state, respectively.

**Figure 5 materials-13-00469-f005:**
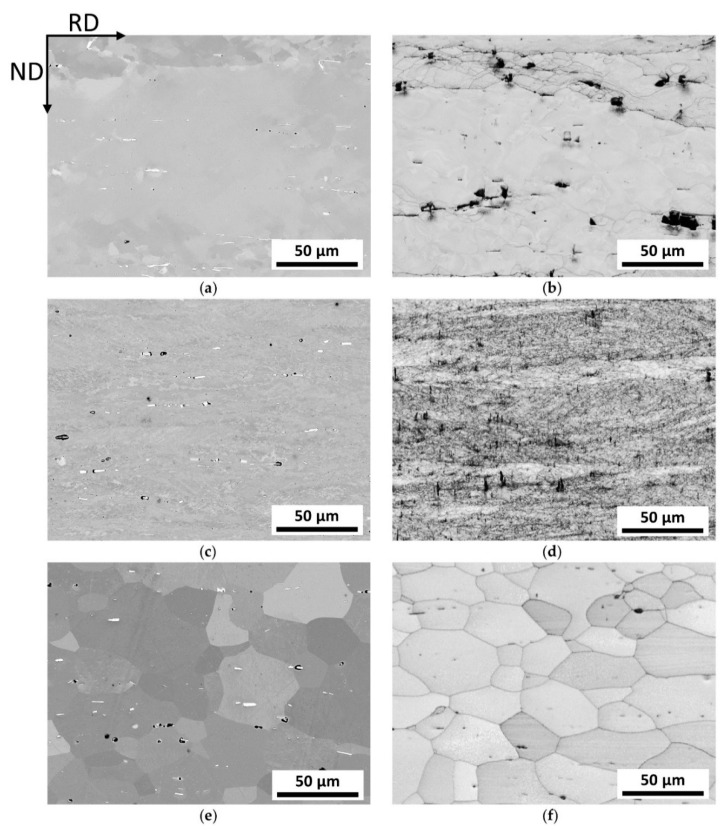
Microstructure evolution of the laboratory processed EN AW-6016 plotted in the TD-plane; (**a**,**c**,**e**) show BSE-images of the (**a**) hot-rolled, (**c**) cold-rolled (after IA), and (**e**) soft annealed state, respectively; (**b**,**d**,**f**) show band contrast maps of the (**b**) hot-rolled, (**d**) cold-rolled (after IA), and (**f**) solution annealed state, respectively.

**Figure 6 materials-13-00469-f006:**
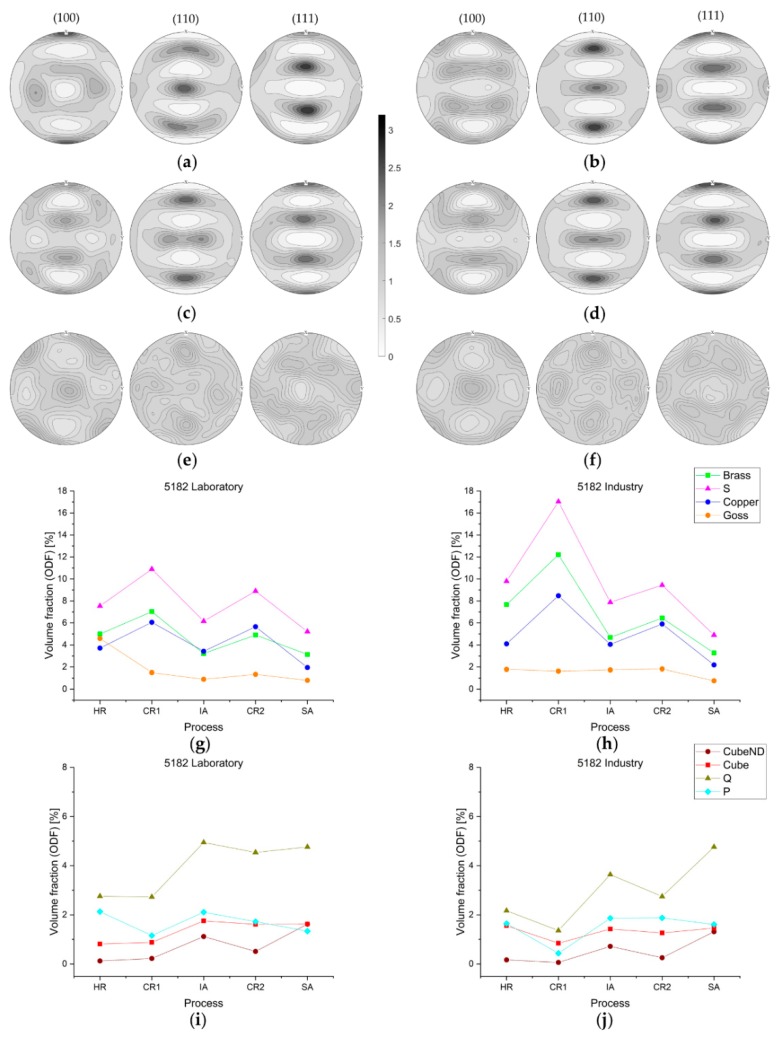
Evolution of texture components within sheet processing for EN AW-5182; (001), (110), and (111) pole figures based on ODF data for (**a**) hot rolling (HR) laboratory, (**b**) HR industry, (**c**) cold rolling (CR)2 laboratory, (**d**) CR2 industry, (**e**) soft annealing (SA) laboratory, and (**f**) SA industry; diagrams (**g**–**j**) show the evolution of specific texture components over the entire processing route.

**Figure 7 materials-13-00469-f007:**
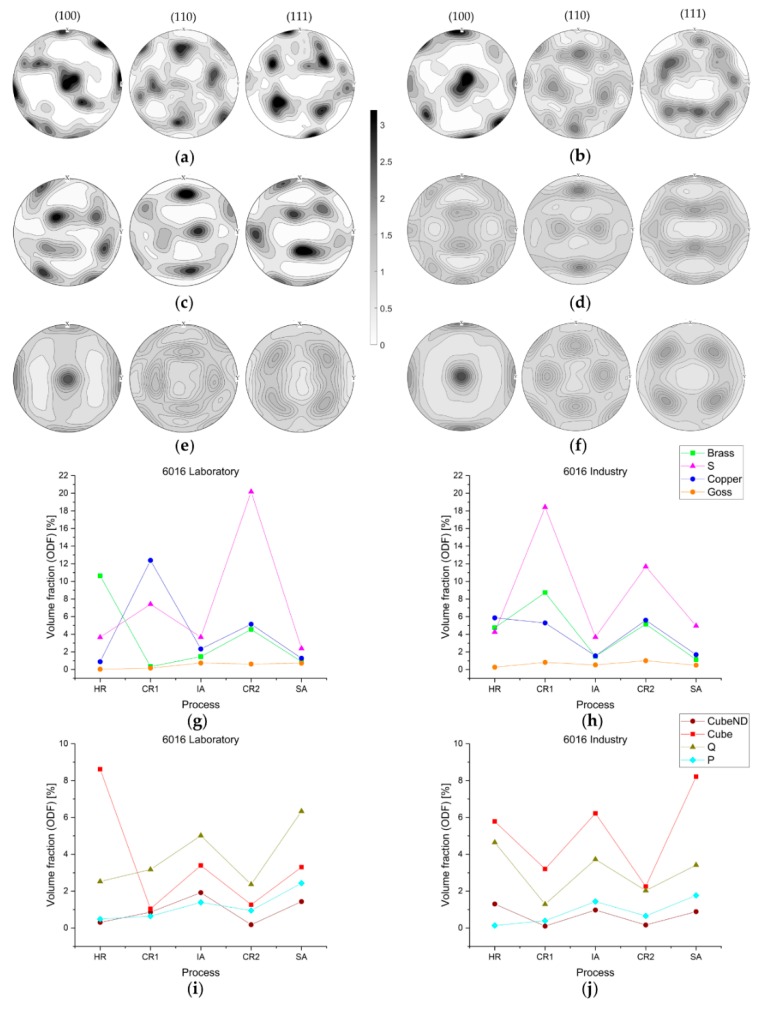
Evolution of texture components within sheet processing for EN AW-6016; (001), (110), and (111) pole figures based on ODF data for (**a**) HR laboratory, (**b**) HR industry, (**c**) CR2 laboratory, (**d**) CR2 industry, (**e**) SA laboratory, and (**f**) SA industry; diagrams (**g**–**j**) show the evolution of specific texture components over the entire processing route.

**Table 1 materials-13-00469-t001:** Chemical composition of the studied alloys EN AW-5182 and EN AW-6016 (in wt %).

Alloy	Mg	Mn	Fe	Si	Al
EN AW-5182	4.83	0.41	0.16	0.09	Bal.
EN AW-6016	0.35	0.07	0.19	1.14	Bal.

**Table 2 materials-13-00469-t002:** Ideal orientations of analyzed texture components and corresponding Euler angles (Bunge convention) [8].

Component		Phi1	Phi	Phi2
Cube	{001}〈100〉	0	0	0
Cube_ND-45_	{001}〈110〉	45	0	0
Goss	{011}〈100〉	0	45	0
Copper	{112}〈111〉	180	21.8	0
		289.5	45	0
S	{123}〈634〉	121	36.7	26.6
		302.3	18.4	0
		301	36.7	26.6
		122.3	18.4	0
Brass	{011}〈211〉	35.3	45	0
		90	144.7	225
Q	{013}〈231〉	239	143.3	206.6
		237.7	161.6	180
		59	143.3	206.6
		57.7	161.6	180
P	{011}〈122〉	90	35.3	45
		0	21.8	360

**Table 3 materials-13-00469-t003:** Microstructural parameter for laboratory and industrially produced EN AW-5182 sheets.

Scale of Production	Direction	HR	CR1	IA	CR2	SA
Laboratory	l_RD_/µm	14 ± 11	22 ± 24	13 ± 9	17 ± 11	22 ± 13
Industry	l_RD_/µm	18 ± 12	12 ± 7	10 ± 4	10 ± 6	20 ± 11

HR: hot-rolling; CR1: cold-rolling before intermediate annealing; IA: intermediate annealing; CR2: cold-rolling after intermediate annealing; SA: soft annealing; l_RD_/µm: linear intercept length in rolling direction.

**Table 4 materials-13-00469-t004:** Microstructural parameter for laboratory and industrially produced EN AW-6016 sheets.

Scale of Production	Direction	HR	CR1	IA	CR2	SA
Laboratory	l_RD_/µm	68 ± 147 *	27 ± 45 *	22 ± 25	14 ± 18	22 ± 15
Industry	l_RD_/µm	73 ± 112 *	16 ± 22 *	37 ± 34	13 ± 11	19 ± 10

* The large error results from a low number of analyzed grains, partially exceeding scan size in the EBSD measurements. HR: hot-rolling; CR1: cold-rolling before intermediate annealing; IA: intermediate annealing; CR2: cold-rolling after intermediate annealing; SA: soft annealing; l_RD_/µm: linear intercept length in rolling direction.

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
