# Peer review of "Evolution of Microstructure and Texture in Laboratory- and Industrial-Scaled Production of Automotive Al-Sheets"

_materials, 2020, doi:10.3390/ma13020469_

Round 1

Reviewer 1 Report

The manuscript presents the evolution of microstructure and texture of Al sheets used for automotive applications, made in two standard Al alloys, EN AW-5182 and EN AW-6016. Laboratory and industrial scale produced materials were investigated at different stages of the production process to highlight the texture and microstructure evolution.

It is suggested to authors to pay attention to the following and to amend accordingly:

For better quality and understanding of the manuscript, some of the Figures in the manuscript need to be improved and adjusted, e.g. in Fig.1, Fig. 6 and Fig. 7 the Miller plane indices (hkl) font size is too small and blur to be clearly seen. The same comment for the evolution of the specific texture components over the entire processing route. The diagrams in Fig. 6 and Fig. 7 are blurred and the colours of the lines are difficult to be distinguish. Page 8, row 286. It is not explained how EDS analysis was used to investigate the formation of primary and secondary Al- (Fe, Mn) intermetallic, as well as the results that could confirm this. Due to EDS limitations, it would be much better if the microstructure features were confirmed by other method, for example by XRD results. The novelties addressed and the original findings should be highlighted. The manuscript writing style should be carefully revised.

Reviewer 2 Report

Line 186 - What is a chemical composition of etchant AlEP2 used in this investigation ? Please provide this information.

Line 268 - "typical inclining to RD (green arrow Figure 4 (b)) " Isnt that slip bands in the grain interiors ?

Line 270 - " The BC-image given in Figure 4 (b) shows high angle grain boundaries (HAGB)... " How authors recognize HAGBs in this image ?

Figure 6 and Figure 7 - Is it possible to put information about main texture components on the pole figures ?

Reviewer 3 Report

Fig. 4 and 5 - Distinguish in the figure caption which letter corresponds to which image (e.g. by adding ", respectively")

Table 3 and 4 - with such high error, I would use only the values without the decimal part.

Table 3 - Why is the LIL after CR increasing in laboratory conditions and decreasing/constant in industry conditions?

l. 320 - Why is the defromation structure stronger in AW-6016?

Round 2

Reviewer 2 Report

The article can be published in a present form.